# Synthetic Lethality Screening Identifies FDA-Approved Drugs that Overcome ATP7B-Mediated Tolerance of Tumor Cells to Cisplatin

**DOI:** 10.3390/cancers12030608

**Published:** 2020-03-06

**Authors:** Marta Mariniello, Raffaella Petruzzelli, Luca G. Wanderlingh, Raffaele La Montagna, Annamaria Carissimo, Francesca Pane, Angela Amoresano, Ekaterina Y. Ilyechova, Michael M. Galagudza, Federico Catalano, Roberta Crispino, Ludmila V. Puchkova, Diego L. Medina, Roman S. Polishchuk

**Affiliations:** 1Telethon Institute of Genetics and Medicine, Pozzuoli, 80078 Naples, Italy; marta.mariniello@uzh.ch (M.M.); r.petruzzelli@tigem.it (R.P.); l.wanderlingh@tigem.it (L.G.W.); lamontagna82@gmail.com (R.L.M.); carissimo@tigem.it (A.C.); f.catalano@tigem.it (F.C.); r.crispino@tigem.it (R.C.); medina@tigem.it (D.L.M.); 2Institute for Applied Mathematics ‘Mauro Picone’, CNR, 80131 Naples, Italy; 3Department of Chemistry, University Federico II, 80126 Naples, Italy; francesca.pane@unina.it (F.P.); angela.amoresano@unina.it (A.A.); 4ITMO University, 197101 Saint Petersburg, Russia; ilichevaey@gmail.com (E.Y.I.); puchkovalv@yandex.ru (L.V.P.); 5Institute of Experimental Medicine, the Russian Academy of the Sciences, 197376 Saint Petersburg, Russia; 6Institute of Experimental Medicine, Almazov National Medical Research Centre, 197101 Saint Pietersburg, Russia; galagoudza@mail.ru; 7Institute of Biosciences and Bioresources, CNR, 80131 Naples, Italy

**Keywords:** cancer, cisplatin resistance, ATP7B, copper transporters, synthetic lethality screening, FDA-approved drugs

## Abstract

Tumor resistance to chemotherapy represents an important challenge in modern oncology. Although platinum (Pt)-based drugs have demonstrated excellent therapeutic potential, their effectiveness in a wide range of tumors is limited by the development of resistance mechanisms. One of these mechanisms includes increased cisplatin sequestration/efflux by the copper-transporting ATPase, ATP7B. However, targeting ATP7B to reduce Pt tolerance in tumors could represent a serious risk because suppression of ATP7B might compromise copper homeostasis, as happens in Wilson disease. To circumvent ATP7B-mediated Pt tolerance we employed a high-throughput screen (HTS) of an FDA/EMA-approved drug library to detect safe therapeutic molecules that promote cisplatin toxicity in the IGROV-CP20 ovarian carcinoma cells, whose resistance significantly relies on ATP7B. Using a synthetic lethality approach, we identified and validated three hits (Tranilast, Telmisartan, and Amphotericin B) that reduced cisplatin resistance. All three drugs induced Pt-mediated DNA damage and inhibited either expression or trafficking of ATP7B in a tumor-specific manner. Global transcriptome analyses showed that Tranilast and Amphotericin B affect expression of genes operating in several pathways that confer tolerance to cisplatin. In the case of Tranilast, these comprised key Pt-transporting proteins, including ATOX1, whose suppression affected ability of ATP7B to traffic in response to cisplatin. In summary, our findings reveal Tranilast, Telmisartan, and Amphotericin B as effective drugs that selectively promote cisplatin toxicity in Pt-resistant ovarian cancer cells and underscore the efficiency of HTS strategy for identification of biosafe compounds, which might be rapidly repurposed to overcome resistance of tumors to Pt-based chemotherapy.

## 1. Introduction

Platinum (Pt)-based drugs have had a major impact in cancer medicine since they display excellent efficacy in the treatment of a wide variety of solid tumors including testicular, head and neck, lung, ovarian, and colon cancers [1]. The mechanism by which these drugs exert their anti-tumor effects is defined by their capacity to form intra-strand DNA adducts that ultimately culminate in tumor cell apoptosis [2]. However, the final outcome of Pt-based therapy is often hampered by the development of drug resistance, which remains a significant obstacle to successful treatment in the clinic [1,2]. Pt resistance usually emerges as a result of the orchestrated action of several pathways, which include cytosolic inactivation of Pt by sulfur-containing molecules like glutathione and metallothionein, activated DNA repair, and reduced Pt accumulation [1,2]. Reduced Pt accumulation depends significantly on changes in the expression and activity of membrane copper transporters (CTR1 and CTR2) that regulate Pt uptake across the plasma membrane [3,4] and the copper transporting ATPases (ATP7A and ATP7B) that drive Pt export from the cell [5,6,7]. In particular, elevated expression of ATP7B has been correlated with a worse outcome of cisplatin chemotherapies in patients with different cancers [8,9].

ATP7B is normally expressed in liver hepatocytes and operates in maintaining copper homeostasis in the body. In response to Cu overload, ATP7B moves from the Golgi to endo-lysosomal structures where it sequesters excess Cu and promotes its excretion into the bile [10,11]. Loss-of-function mutations prevent ATP7B from sequestering Cu and its removal from cells leading to severe Cu toxicosis, which is known as Wilson disease [12]. 

A similar ATP7B-dependent mechanism has been proposed to reduce Pt toxicity in tumor cells. It was hypothesized that ATP7B promotes sequestration of Pt within intracellular post-Golgi vesicles and/or excretion of Pt at the cell surface [11,12,13,14]. In agreement with this hypothesis, extensive relocation of ATP7B from the Golgi to vesicular compartments was observed in Pt-resistant tumor cells [15]. In addition, ATP7B has been proposed to chelate Pt via its metal-binding domains, thus reducing Pt toxicity [16]. In this context, circumventing ATP7B-mediated resistance emerges as an attractive strategy to increase the efficacy of Pt-based cancer therapies.

A growing body of evidence suggests that the inhibition of single pathways that support cisplatin resistance fails to restore sensitivity to normal levels and that the treatments for overcoming resistance have to target distinct mechanisms [2]. Therefore, we reasoned that screening for FDA/EMA-approved drugs might offer an effective shortcut to identify safe molecules that could reduce tolerance to cisplatin in tumor cells. Considering that the safety profiles of such drugs are well characterized, their repurposing for other therapeutic uses significantly reduces the risk of failure due to eventual adverse events in patients [17]. Using a synthetic lethality screen, we identified and validated three hit drugs that significantly reduced ATP7B-mediated tolerance to cisplatin in ovarian cancer cells. The drugs act via several mechanisms including the tumor-specific suppression of ATP7B expression and trafficking and the downregulation of large cohorts of genes that belong to cisplatin-resistance pathways including DNA repair, protein quality control, and autophagy [18,19,20,21]. Our findings underscore the effectiveness of an unbiased screening approach for drug repurposing studies and pinpoint bio-safe drug candidates to combat ATP7B-mediated resistance of tumors to cisplatin.

## 2. Results

### 2.1. IGROV-CP20 Cells Represent a Robust System to Study Cisplatin Resistance

Considering that our study was focused on screening for drugs promoting cisplatin toxicity, we first selected the appropriate cell line to perform HTS. To this end we evaluated the response to cisplatin of two well-known ovarian cancer cell lines, IGROV-CP20 and A2780-CP20, whose tolerance to Pt drugs has been reported to significantly rely on ATP7B [14,22]. Initially, resistance to cisplatin was investigated in the Pt-resistant IGROV-CP20 and the parental Pt-sensitive IGROV cell lines treated with increasing concentrations of cisplatin. IGROV-CP20 exhibited significantly higher viability at different cisplatin concentrations than the parental IGROV line (Figure 1A). In contrast, the difference in survival of resistant A2780-CP20 and sensitive A2780 cells was less evident (Appendix A), indicating the IGROV-CP20 line as a more robust cell system for the Pt resistance study with HTS. Of the tested concentrations, 50 µM cisplatin resulted in the highest difference in survival between IGROV-CP20 and IGROV cells (Figure 1A). Therefore, this regimen of cisplatin treatment in IGROV-CP20 cells was further employed for drug screening.

To ensure that ATP7B provides a significant contribution to cisplatin resistance in IGROV-CP20 cells, we first assessed the expression levels of ATP7B. Western blot revealed higher levels of ATP7B in IGROV-CP20 cells compared to the parental IGROV line (Figure 1B). In parallel, elevated ATP7B expression in IGROV-CP20 cells was detected by both qRT-PCR and immunofluorescence (Figure 1C,D). Finally, ATP7B silencing (Appendix A) resulted in a two-fold decrease in the viability of IGROV-CP20 cells upon cisplatin treatment (Figure 1E), indicating that Pt-resistance of IGROV-CP20 cells relies on ATP7B in a significant manner.

These findings did not formally rule out the possibility that other copper transporters such as ATP7A or CTR1 may contribute to the cisplatin tolerance of IGROV-CP20 cells. Indeed, ICROV-CP20 cells exhibited slightly higher levels of ATP7A protein and mRNA compared to parental cells (Figure 1F,G). Considering that elevated ATP7A expression confers tolerance to cisplatin in a number of tumors [23,24], we decided to check whether this is a case in IGROV-CP20 cells. However, RNAi-mediated suppression of ATP7A (Appendix A) did not result in any significant decrease in the resistance of the cells to cisplatin (Figure 1H). Next, we checked CTR1 expression levels in IGROP-CP20 cells because reduced expression of this transporter has been associated with limited cisplatin uptake and, as a consequence, with higher Pt tolerance [3,4]. However, we found that CTR1 expression is higher in the IGROV-CP20 cell line compared to the parental line (Figure 1I,J). Taken together these findings argue against a substantial involvement of either ATP7A or CTR1 in cisplatin tolerance in IGROV-CP20 cells and confirm a key contribution of ATP7B to Pt resistance. 

### 2.2. Synthetic Lethality Screening Identified FDA-Approved Drugs Accelerating Pt-Mediated Death of Resistant IGROV-CP20 Cells

The HTS strategy to identify FDA-approved drugs reducing the resistance of IGROV-CP20 cells to cisplatin was based on the principle of synthetic lethality. The Prestwick Chemical Library of 1280 compounds was dispensed at a concentration of 10 μM in duplicate on the IGROVCP20 cells in 384-well formats. An MTT colorimetric assay (see Section 4) was used as readout for viability of IGROVC-P20 cells upon drug and cisplatin treatments. As shown in the heat maps of multi-well plates (Figure 2A), this assay generated a different degree of MTT signal. Parental IGROV cells were used as a positive control to test the impact of cisplatin on cell survival and were plated in the first column of the 384-well plates, while the last two columns were left blank as a negative control for the assay (Figure 2A). After the addition of the library compounds, one plate from duplicate was treated with 50 µM of cisplatin for 24 h while the other was left without cisplatin to test drug toxicity. This allowed us to exclude those drugs that are toxic even without cisplatin (Figure 2A, black boxes), while the drugs promoting cell death only in combination with cisplatin were considered as positive hits (Figure 2A, red boxes). 

Several controls were performed to ensure the assay reproducibility for HTS. First, the assay robustness was estimated by calculating the Z-score factor, which defines the reproducibility and suitability of the assay for HTS [25]. To this end, wells containing IGROV and IGROV-CP20 cells were compared after 24 h exposure to cisplatin. Evaluation of the MTT signal resulted in a Z’-factor value of 0.279 across the two cell lines (Appendix A), thereby indicating that the differences in cell response is sufficient for HTS [25]. Secondly, a plate uniformity test was performed to exclude “drift and edge effect” in the plate as well as any other systematic source of variability among the samples [25]. Appendix A shows that the plates exhibited good uniformity during the screening and no drift and edge effects were detected.

For the HTS, each component of the library was analyzed in three independent experiments with or without addition of cisplatin. The correlation between three replicates was calculated using a Pearson’s coefficient that was consistently robust across the experiments (Appendix A), indicating that in each replicate the drugs had a similar impact (as is desirable for the overall success of HTS screening) [25]. Positive hits were identified on the basis of the decrease in cell viability. The arbitrary cut-off threshold for hit detection was set at a level of 1.5 standard deviations below the average viability of resistant cells in cisplatin. However only drugs without own toxicity were selected for further validation (Figure 2B, red dots). Overall, the screen revealed eight FDA candidates that induced a strong reduction in Pt-resistance of IGROV-CP20 cells (Figure 2B,C). The presence of Amphotericin B among the hits underscores the efficiency of the screening because this drug is well known for its ability to reduce Pt tolerance [26]. The hit compounds were then tested in dose-response experiments using different concentrations of each drug (ranging from 0.1 to 30 μM) in combination with cisplatin (Figure 2D; Appendix A). The dose-dependent curves showed that some concentrations of the tested drugs reduced tolerance of the cells to cisplatin (Appendix A). However, only Tranilast, Telmisartan, and Amphotericin B exhibited a correlation between drug concentration and degree of cell survival in cisplatin (Figure 2D). Therefore, these three drugs were considered as robust hits for further validation. Notably, a few drugs (Moxonidine, Gestrinone, and Alosetron hydrochloride) promoted tolerance of IGROV-CP20 cells to cisplatin (Appendix A) and, thus, their use might represent a risk during the course of cisplatin chemotherapy.

### 2.3. Validation of Hit Impact on Resistance of IGROV-CP20 Cells to Cisplatin

Several assays were employed to validate the efficacy of each HTS hit. Selective live/dead cell staining showed that the toxicity of cisplatin for IGROV-CP20 cells was very limited (Figure 3A,B). However, the level of cell mortality substantially increased when cisplatin treatment was supplemented with Tranilast, Telmisartant, or Amphotericin B (Figure 3A,B). The ability of the drug hits to promote cisplatin toxicity was also confirmed on the other ATP7B-mediated resistant tumor cell line A2780-CP20 (Appendix A). However, Amphotericin B appeared to be quite toxic even alone in this cell type. Notably, none of the drugs were able to further increase Pt-mediated death in Pt-sensitive IGROV cells (Appendix A).

At this point it was relevant to verify the potential of each drug to promote the formation of Pt DNA adducts in the resistant cells. To this end we employed a dot immuno-blot assay with a specific antibody capable of recognizing DNA adducts (see Section 4). After cisplatin treatment Pt-sensitive IGROV cells exhibited a significantly higher adduct-associated signal compared to the resistant IGROV-CP20 cells (Figure 3C,D). Addition of hit drugs in combination with cisplatin significantly increased the adduct signal of spotted DNA compared to cisplatin alone (Figure 3C,D) even though only Amphotericin B elevated overall intracellular levels of Pt (Figure 3E). Moreover, the drugs increased the abundance of adducts in IGROV-CP20 cells to the levels similar to those observed in the Pt-sensitive parental cell line (Figure 3C,D), thereby indicating a significant potential of the drug hits to promote Pt-mediated damage of DNA in resistant cells.

Finally, the impact of all three hits on intracellular Pt accumulation was investigated using ICP-MS. In control experiments we found that exposure of sensitive IGROV cells to cisplatin resulted in a significant increase in the intracellular Pt levels, while Pt remained low in the resistant IGROV-CP20 cells even after 24 h incubation with cisplatin (Figure 3E). Interestingly, only Amphotericin B induced a significant accumulation of Pt in IGROV-CP20 cells. It has been reported that Amphotericin B acts as an ionophore that forms pores in cell membrane, thereby, facilitating cisplatin influx [27]. In contrast, the addition of Tranilast or Telmisartan in combination with cisplatin did not cause any significant increase in Pt levels (Figure 3E). This prompted us to further investigate the mechanism through which these drugs amplify the sensitivity of resistant tumor cells to cisplatin.

### 2.4. FDA-Approved Drug Hits Affect Expression and Trafficking of Copper Transporters in Pt-Resistant Tumor Cells

Initially, we evaluated the impact of the drug hits on the expression of ATP7B in Pt-treated IGROV-CP20 cells using qRT-PCR and Western blot. Although none of the drugs induced a significant change in ATP7B transcripts (Figure 4A), Western blot showed that Amphotericin B, but not the other drugs, caused a substantial decrease in ATP7B protein levels (Figure 4B). This indicated that in addition to known mechanisms [27] lower ATP7B protein levels might contribute to Pt accumulation in Amphotericin B-treated cells. The expression of other Pt transporters, ATP7A and CTR1, was also analyzed in Pt-resistant cells. We found that Amphotericin B treatment caused a decrease of ATP7A mRNA and very strong depletion of the ATP7A protein (Figure 4C,D). Tranilast- and Termisaltan-treated cells exhibited a slight reduction in ATP7A protein but not in mRNA (Figure 4C,D). Finally, none of the drugs altered the CTR1 transcript levels in Pt-treated cells while CTR1 protein levels were very slightly lowered by Tranilast and Telmisartan (Figure 4E,F).

The lack of a striking impact of Tranilast and Telmisartan on the expression of ATP7B (and other Pt transporters) prompted us to examine the intracellular localization of ATP7B in IGROV-CP20 cells. Predictably, we found that addition of cisplatin stimulated export of ATP7B from the Golgi (Figure 4G), as this process is apparently required for proper Pt sequestration/efflux [12,13,14,15]. However, pretreatment of IGROV-CP20 cells with Tranilast, Telmisartant, or Amphotericin B inhibited Pt-mediated export of ATP7B from the Golgi (Figure 4G). This was particularly evident for Tranilast and Telmisartan treatments, in which a significant increase in ATP7B retention in the Golgi region was detected (Figure 4G,H). These results suggest that the drugs could have an impact on ATP7B trafficking, thereby inhibiting ATP7B-mediated sequestration and efflux of Pt and, thus, favoring tumor cell death. 

However, in the context of eventual clinical use, the inhibitory impact of these drugs on ATP7B trafficking might present a risk of toxic copper accumulation in the liver where ATP7B trafficking is needed for sequestration of excess copper and its further excretion to the bile [12]. Therefore, we tested whether Tranilast, Telmisartan, or Amphotericin B alter copper-dependent ATP7B trafficking in hepatic HepG2 cells, but none of the drugs inhibited copper-induced export of ATP7B from the Golgi (Appendix A). In line with this, all three drugs failed to stimulate cisplatin toxicity in HepG2 cells (Appendix A) indicating that they act though a mechanism that is likely to be specific for Pt-resistant ovarian tumor cells.

### 2.5. The Drug Hits Affect Expression of Genes Operating in Different Pt-Resistance Pathways

Considering that cisplatin tolerance usually relies on several different mechanisms, even in a single cell type [2], we sought to investigate whether Tranilast, Telmisartan, or Amphotericin B affect Pt-resistance pathways that do not involve ATP7B. To test this, QuantSeq mRNA technology was employed to analyze the global transcriptional response to each drug. Like qRT-PCR, QuantSeq 3’ mRNA sequencing allows very accurate measurements of quantitative differences in gene expression [28].

Thus, gene expression values in IGROV-CP20 cells, which were treated with cisplatin alone, were compared to those in the cells treated with a combination of cisplatin and each drug hit (Gene Expression Omnibus accession number GSE134029). QuantSeq analysis showed that all three drugs modulated gene expression in cisplatin-treated cells (Figure 5A; Appendix A). To determine the biological pathways affected by drugs in cisplatin-treated cells, gene ontology (GO) enrichments were calculated (see Section 4). GO analysis of Tranilast- and Amphotericin B-treated cells revealed several statistically significant GO enrichments that exhibited a similar trend in several groups of downregulated genes (Figure 5B; Appendix A). These include DNA repair, protein quality control, ubiquitination and folding, macroautophagy, and NF-κB signaling, all of which have been reported to promote cisplatin resistance [18,19,20,29]. By contrast, the transcriptional response of Pt-treated IGROV-CP20 cells to Telmisartan exhibited a different trend. Only some protein quality control pathways were downregulated in common with Tranilast and Amphotericin B, while we found Telmisartan to suppress genes driving the response to hypoxia (Figure 5B; Appendix A), whose involvement in cisplatin resistance has been reported [30].

Importantly, Tranilast-treated samples displayed a significant downregulation of genes encoding copper ion-binding proteins (Figure 5B). This cohort of genes comprised ATOX1, COX17, and SOD1 (Figure 6A), which are involved in copper and platinum distribution towards different cellular destinations such as the secretory system, mitochondria, and free radical detoxification pathways, respectively [31,32,33]. Thus, by affecting ATOX1, COX17, and SOD1, Tranilast might inhibit delivery of Pt to these pathways, thereby favoring its transport to the nucleus, where Pt promotes DNA damage resulting in cell death (Figure 6B).

### 2.6. Tranilast Affects ATP7B Trafficking in Pt-Resistant Tumor Cells via Downregulation of ATOX1

Among the copper-binding genes that were downregulated by Tranilast, we decided to focus on *ATOX1*, which has been suggested to promote ATP7B-mediated Pt resistance [34]. qRT-PCR revealed IGROV-CP20 cells to activate ATOX1 expression in response to cisplatin, while Tranilast significantly attenuated Pt-mediated induction of ATOX1 (Figure 6C). This transcriptional response of ATOX1 to Tranilast mirrored that observed with RNAseq, and, thus, showed that our transcriptomics data provide meaningful information regarding the mechanism of action of the drug hits. Importantly, we did not find any significant impact of either cisplatin or Tranilast on ATOX1 expression in hepatic HepG2 cells (Appendix A), indicating that ATOX1 might be targeted by Tranilast in a safe tumor-specific manner. To test this hypothesis, we sought to investigate whether overexpression of ATOX1 is able to overcome the impact of Tranilast by recovering (at least partially) the resistance of IGROV-CP20 cells to cisplatin. 

We reasoned that overexpressing exogenous ATOX1 under the generic CMV promoter might allow IGROV-CP20 cells to circumvent the inhibitory impact of Tranilast on transcription of endogenous ATOX1. To this end, the cells were transfected with ATOX1-FLAG plasmid [35] and incubated with cisplatin alone or in combination with Tranilast. Initially, we analyzed the impact of ATOX1 overexpression on Pt-dependent trafficking of ATP7B in Tranilast-treated cells. Confocal microscopy revealed that ATOX1-overexpressing cells supported Pt-induced relocation of ATP7B from the Golgi to the cell periphery even in the presence of Tranilast (Figure 6D). In contrast, Tranilast still inhibited Pt-dependent export of ATP7B from the Golgi in the cells that did not overexpress ATOX1 (Figure 6D). Correspondingly, the MTT assay confirmed that the overexpression of ATOX1 blocked the ability of Tranilast to promote cisplatin toxicity in IGROV-CP20 cells (Figure 6E). Furthermore, dot blot revealed that ATOX1 overexpression significantly reduced the capacity of Tranilast to induce formation of DNA-platinum adducts (Figure 6F). These findings indicate that ATOX1 contributes to cisplatin resistance in ovarian cancer cells, while its downregulation with Tranilast promotes Pt-mediated cell death.

## 3. Discussion

Although Pt-based drugs have demonstrated excellent therapeutic potential, their effectiveness in a wide range of tumors is limited by the development of resistance mechanisms including increased Pt sequestration/efflux by ATP7B [1,2]. In this context ATP7B emerges as an attractive target to combat Pt resistance. Indeed, silencing ATP7B has been demonstrated to reduce tolerance of tumor cells to Pt-based drugs in vitro and in vivo [22]. However, the ATP7B RNAi strategy has never been advanced to therapy due to potential risks of compromising Cu homeostasis in patients. Considering that ATP7B suppression leads to rapid accumulation of hepatic copper and liver damage [12], ATP7B-activity in Pt-resistant tumors should be targeted or circumvented in a safe manner to avoid hepatic complications. In this context, we reasoned that identification of bio-safe FDA/EMA-approved drugs might help to achieve this objective. 

Here, using synthetic lethality screening, we detected and validated three FDA-approved drugs, Tranilast, Telmisartan, and Amphotericin B, which promote Pt-mediated death in ovarian cancer cells, whose resistance to cisplatin relies on ATP7B in a significant manner. All three drugs enhanced cisplatin toxicity via stimulation of Pt-DNA adduct formation. Notably, the efficacy of Tranilast and Telmisartan was confirmed in another cisplatin-resistant ovarian cancer cell line, A2870-CP20. However, these drug hits did not enhance cisplatin toxicity in the parental Pt-sensitive IGROV cells supporting the notion that resistant cells develop specific tolerance mechanisms [1,2] that might be targeted in a selective way. 

Despite the similar impact on cisplatin tolerance, the drug hits exhibited a clear difference in their ability to promote accumulation of Pt in resistant cells. Only Amphotericin B treatment resulted in a significant increase in intracellular Pt levels. In addition to the known impact of Amphotericin B on membrane permeability [27], substantial reduction in ATP7B protein levels might contribute to Pt buildup in Amphotericin B-treated cells. 

In contrast to Amphotericin B, Tranilast and Telmisartan did not accelerate Pt accumulation but were still able to facilitate DNA adduct formation. It is tempting to hypothesize that both drugs affect the intracellular distribution of Pt thereby favoring delivery of the metal to the nucleus, where it damages DNA. Indeed, both Tranilast and Telmisartan inhibited export of ATP7B from the Golgi in response to cisplatin. This trafficking event is apparently needed for delivery of ATP7B to the peripheral structures [13,15], where reduced pH favors the metal transporting activity of ATP7B [36] and, hence, sequestration of Pt in the lumen of these organelles. Therefore, failure of ATP7B trafficking in Tranilast- and Telmisartan-treated cells favors toxic accumulation of Pt in other cell compartments and increases its chances to arrive in the nucleus and damage the DNA. Interestingly, some ATP7B retention was also observed in the case of Amphotericin B treatment of ICROV-CP20 cells but to a lesser degree than observed with the other drugs. 

The mechanisms by which the drug hits inhibit ATP7B trafficking remain unclear except for Tranilast, which reduces ATOX1 expression in resistant tumor cells. Elevated expression of ATOX1 has been associated with tolerance to cisplatin in several tumor cell lines [34]. In line with this notion, we found that Pt-resistant IGROV-CP20 cells upregulate ATOX1 in response to cisplatin. ATOX1 operates as a specific chaperone, which ferries incoming Cu or Pt to ATP7B thereby stimulating metal-dependent trafficking of ATP7B [32]. Tranilast-mediated suppression of ATOX1 could reduce the supply of Pt to ATP7B, thus inhibiting ATP7B trafficking to post-Golgi compartments, which operate in Pt sequestration/efflux. In line with our observations, silencing ATOX1 has been reported to reduce tolerance to cisplatin in tumor cells [37]. Importantly, we found that Tranilast did not affect ATOX1 expression, ATP7B trafficking, or cisplatin toxicity in hepatic cells, where ATP7B is required to maintain Cu homeostasis. This indicates that Tranilast targets ATP7B-mediated resistance mechanism in a tumor-specific manner, which would strongly reduce the risk of side effects and adverse events upon therapeutic use.

Although ATP7B expression and trafficking provided a significant contribution to cisplatin tolerance in ovarian cancer cells tested, it is likely that other mechanisms are involved in Pt-resistance because cisplatin treatment usually causes complex and multifactorial cellular responses [2]. Notably, all three FDA-approved hits promoted cisplatin toxicity in resistant IGROV-CP20 cells to the levels observed in parental sensitive IGROV cells (see Figure 2C and Figure 3A–D). Such efficiency could be achieved only in the case when several mechanisms of cisplatin resistance are suppressed by the treatment [2]. Thus, to understand how these drugs affect responses of tumor cells to cisplatin, we used a systems biology approach and investigated drug-induced changes to the transcriptome. GO analysis revealed that among the Pt-resistance mechanisms, only those associated with a reduction of protein quality control [20] were inhibited by all three drugs. The overlap between the transcriptional responses to Tranilast and Amphotericin B was wider and included suppression of such Pt resistance pathways as DNA repair [21] and anti-apoptotic NF-κB signaling [18]. In addition, we noted that both drugs reduced expression of autophagy genes. Autophagy has recently emerged as an important mechanism for Cu and Pt detoxification [19,38] and its role in resistance of tumor cells to cisplatin has been documented [19]. 

However, the impact of Tranilast on the transcriptome of resistant cells exhibited a particular feature that consisted of downregulation of several genes regulating Cu metabolism. This finding was of particular interest because Cu and Pt are frequently handled by the same intracellular pathways that also operate in cisplatin resistance [12,14]. The cohort of Tranilast-sensitive transcripts included ATOX1, COX17, and SOD1 that normally supply Cu to biosynthetic, mitochondrial, and anti-oxidant pathways, respectively [31,32,33]. Pt binding to ATOX1, COX17, and SOD1 has been documented [34,39,40,41]. Moreover, an increase in SOD1 and ATOX1 expression has been suggested to confer cisplatin resistance to tumor cells [34,42]. Thus, reduced expression of these three genes in Tranilast-treated cells should result in increased amounts of Pt directed towards other cellular destinations, including the nucleus and DNA (see scheme in Figure 6B). Indeed, we found that Tranilast treatment strongly promoted Pt-induced DNA damage in resistant tumor cells. Suppression of ATOX1 might also help to combat tumor growth through additional mechanisms. As mentioned above, an ATOX1 deficit inhibits ATP7B trafficking and hence its ability to sequester/efflux toxic Pt.

Regardless of the mechanistic similarities and differences between all three identified FDA-approved drugs, they share a number of clinically important features. First, they efficiently circumvent cisplatin resistance in ovarian cancer cells. Thus, considering the known biosafety profiles of these drugs, their translation to clinical use might be relatively rapid. Tranilast inhibits the production of interleukin-6 and is normally used as an anti-allergic drug, which is well tolerated by most patients at very substantial doses [43]. Notably, a new compound based on cisplatin and Tranilast was developed recently and exhibited excellent cytotoxicity in tumor explants [44]. Telmisartan is also quite a safe agent, which operates as an angiotensin II receptor antagonist to treat hypertension [45]. In contrast, Amphotericin B appears to be the least safe. In our hands, it was very toxic alone (i.e., without cisplatin) in the A2780-CP20 cell line. Furthermore, Amphotericin B is known for extensive nephrotoxicity, which limits its clinical use as cisplatin potentiator [46] although some strategies to reduce its toxicity were recently developed [47]. 

Interestingly, Tranilast and Amphotericin B were reported to promote Pt toxicity in a number of tumor cell lines derived from different organs [48,49]. Therefore, their use might be expanded beyond ovarian cancer. Second, none of the three drugs potentiated cisplatin toxicity in the parental ovarian cancer cells, which do not manifest any significant resistance to Pt-based drugs. This indicates that a combination of Tranilast, Telmisartan, or Amphotericin B with cisplatin should be employed only to treat resistant tumors, while the use of these drugs would be worthless in patients that respond efficiently to cisplatin therapy. Finally, despite the ability to suppress Pt-resistance mechanisms including ATP7B trafficking, none of the three drugs stimulate Pt toxicity in hepatic cells. This indicates that these drugs target tumor-specific mechanisms of cisplatin resistance in a highly selective manner and, hence, do not pose any significant risk of promoting Pt toxicity or compromising Cu homeostasis in the liver, thereby reducing the risks of adverse events in eventual clinical use.

## 4. Materials and Methods 

### 4.1. Antibodies and Other Reagents

The full list of antibodies, small interfering RNAs (siRNAs), and primers is provided in the Appendix A.

### 4.2. Cell Culture

Cisplatin-sensitive human ovarian cancer IGROV and A2780 cells and the corresponding cisplatin resistant lines IGROV-CP20 and A2780-CP20 were obtained from Dr. A Sood (University of Texas, MD Anderson Cancer Center). The cells were grown in RPMI supplemented with 15% FBS, 2 mM L-glutamine, and 1% penicillin and streptomycin. HepG2 cells were grown in DMEM supplemented with 10% FCS, 2 mM L-glutamine, and 1% penicillin and streptomycin.

### 4.3. RNA Interference

Small interfering RNAs (siRNA) targeting ATP7B and ATP7A were purchased from Sigma-Aldrich (St. Louis, MO, USA). Scrambled siRNAs were used as a negative control. IGROV-CP20 cells were transfected with siRNA using Lipofectamine RNAiMAX reagent (Invitrogen, Paisley, UK). After 48 h of interference, the cells were treated with 50 µM cisplatin for 24 h and the MTT cell viability assay (see below) was then performed. SiRNA-treated cells were then prepared for quantitative real time PCR to analyze the silencing efficiency.

### 4.4. RNA Preparation and Quantitative Real Time PCR (qRT-PCR)

To evaluate the mRNAs expression levels of ATP7B, ATP7A, CTR1, and ATOX1 genes, total RNAs were extracted with RNeasy Protect Mini kits (Qiagen, Milan, Italy) from treated and control cells. RNAs were reverse-transcribed using QuantiTect Reverse Transcription kits (Qiagen). qRT-PCR experiments were performed using Light Cycler 480 Syber Green I Master Mix (Roche, Basel, Switzerland) for cDNA amplification and the qRT-PCR was carried out in a LightCycler 480 II (Roche) for signal detection. RT-PCR results were analyzed using the comparative CT (threshold cycle) method, normalized against the housekeeping gene β-actin.

### 4.5. Immunofluorescence

A detailed description of the immunofluorescence protocol is provided in the Appendix A. To evaluate the amount of ATP7B in the Golgi, control, and treated IGROV or IGROV-CP20 cells were immuno-labeled for ATP7B and Golgin 97. Golgin 97 staining was used with the ROI manager tool of ImageJ software to generate a mask for the Golgi region in each cell. The mean pixel intensity of the ATP7B signal was measured in the Golgi region of each cell using the Measure tool of ImageJ software and reported as arbitrary units (au).

### 4.6. Western Blot 

Immuno-blotting was performed as described in the Appendix A. Primary antibodies against ATP7B, ATP7A, CTR1, GAPDH, and α-tubulin were used to detect the corresponding proteins.

### 4.7. MTT Cell Viability Assay

Viability of IGROV, IGROV-CP20, A2780, A2780-CP20, and HepG2 cells was determined by measuring their ability to reduce the tetrazolium salt (MTT) to a formazan. The cells were plated in 96-well plates and allowed to adhere overnight. Then the cells were exposed to cisplatin and/or other drugs. After treatment with drugs, the cells were incubated with MTT (Invitrogen). Incubation was stopped using 100 μL of a solution with 25% aqueous ammonia (Sigma) in DMSO (Sigma). Absorbance in each well was recorded at 540 nm in a multi-well plate reader (Synergy/neo, Biotek, Bad Friedrichshall, Germany). The results were normalized to the absorbance value in untreated cells (considered to be 100% viable) and expressed as % of viability.

### 4.8. High-Throughput Screening (HTS)

The overall strategy of HTS is described in the results. For HTS screening, IGROV and IGROV-CP20 cells were plated in 384-well plates. The Prestwick Chemical library of 1280 FDA-approved drugs was screened for compounds that promote cisplatin toxicity in Pt-resistant IGROV-CP20 cells. Triplicates of each drug (at 10 µM concentration) were tested alone for 48 h to evaluate drug toxicity. In parallel sets of plates, each drug was used first alone for 24 h and then combined with 50 µM cisplatin for another 24 h. The STAR-let liquid handling system (Hamilton, Reno, NV, USA) was used to dispense drugs and cisplatin into individual wells. Wells containing Pt-sensitive IGROV cells were used as controls to evaluate the extent of cisplatin toxicity. After treatment the cell viability was evaluated using the MTT assay. The drugs that reduced cell viability only in combination with cisplatin were considered as hits. Overall quality of the screening was controlled through plate uniformity test and correlation between three replicates.

### 4.9. Metal Content Determination by Inductively Coupled Plasma Mass Spectrometry (ICP-MS)

Pt in cell lysates from control and treated cells was analyzed by ICP-MS after wet washing of the samples with nitric acid (Appendix A). Pt concentration was calculated by interpolation under the calibration curve. All values of Pt concentration were for number of cells in each specimen. All analyses were performed in triplicate. 

### 4.10. DNA Adduct Evaluation by Dot Blot

To detect the amount of DNA adducts in control and treated cells, the DNA was extracted from the cells using Quick-DNA Miniprep Plus Kit (Zymo Research, Irvine, CA, USA). Equal amounts of DNA from each specimen were spotted on a piece of Nytran N nylon blotting membrane (GE Healthcare Life Sciences, Marlborough, MA, USA). The membrane was incubated with a primary antibody that recognized Pt-induced DNA adducts (ICR4, Merck, Millipore), then washed with TBST and incubated with an HRP-conjugated secondary antibody according to the manufacturer’s instructions. Chemiluminescent signals were captured using Chemidoc Amersham Imager 600 and quantified using ImageJ software. 

### 4.11. QuantSeq 3’ mRNA Sequencing and Gene Ontology Enrichment Analysis

Three biological replicates of IGROV-CP20 were treated with 50 µM cisplatin directly or after 24 h incubation with 10 µM Tranilast, Amphotericin B, or Telmisartan. The impact of each drug on the transcriptional response to cisplatin was analyzed using QuantSeq 3’ mRNA sequencing. Total RNA was extracted from treated cells using the RNeasy Mini Kit (Qiagen). RNA extracted from the cells treated with cisplatin alone was used as a control. RNA samples were used as templates to prepare corresponding DNA libraries with QuantSeq 3’ mRNA-Seq Library prep kit (Lexogen, Vienna, Austria). Amplified cDNA fragments were sequenced in single-end mode using the NextSeq500 (Illumina, San Diego, CA, USA) with a read length of 75 base pairs. Analysis of the sequence reads and gene ontology (GO) enrichment analysis of transcripts is described in the Appendix A.

### 4.12. ATOX-1 Transfection

For transfection, DNA from the Flag-tagged ATOX-1 plasmid (provided by Prof. T. Fukai) was reverse transfected with Opti-MEM and TransIT^®^-LT1 Transfection Reagent (Mirus Bio LLC, USA) according to manufacturer’s instructions. Two days after transfection, the cells were pretreated with Tranilast (10 µM) for 24 h and then treated with 50 µM cisplatin. After treatment, ATP7B trafficking was analyzed by immunofluorescence and dot immuno-blot analysis was performed to detect the cis-platinum DNA adduct formation. Results shown are from independent transfection experiments, each performed in triplicate.

### 4.13. Live/Dead Fluorescence Cytotoxicity Assay 

Control and treated IGROV or IGROV-CP20 cells were gently washed with PBS. The live/dead reagents (Invitrogen) were dissolved in sterile PBS. Live/dead solution was added to the cells for 30 min to generate a green fluorescent signal in live cells and a red signal in dead cells. The labeled cells were viewed under the ZEISS Axio Observer.Z1 APOTOME fluorescence microscope using FITC and RFP filters to count live/dead cells in the treated and control specimens. Quantification was done in 10 fields for each condition and the proportion of live cells was calculated as % of total cells in the specimen.

### 4.14. Statistical Analysis

Data are expressed as mean ± standard deviation (or standard error where indicated), collected from multiple independent experiments performed on different days. Statistical significance for all data, except the QuantSeq and bioinformatics analyses (Appendix A), was computed using Student’s *t*-test or one-way ANOVA (for all figures, * *p* < 0.05, ** *p* < 0.01, and *** *p* < 0.001 indicate statistical significance). 

## 5. Conclusions

In conclusion, our study demonstrates that HTS emerges as an effective approach that might be used to detect candidate drugs for repurposing in oncological use. Using this strategy, we identified FDA-approved drugs that overcome resistance to cisplatin in ovarian cancer cells and characterized several ATP7B-dependent/independent molecular mechanisms behind the drug impacts. We believe that identification of these drugs will help to accelerate development of new strategies to combat cancer resistance to cisplatin therapy, which still represents an important task in modern oncology.

## Figures and Tables

**Figure 1 cancers-12-00608-f001:**
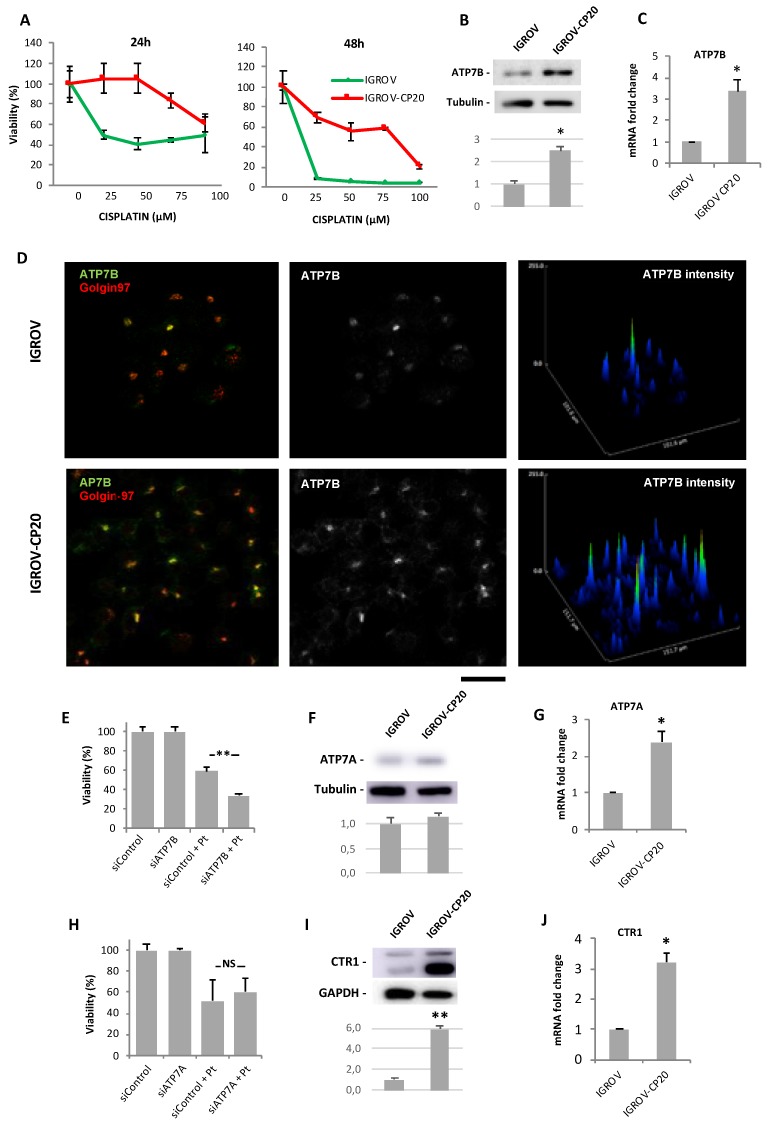
Cisplatin resistance of IGROV-CP20 cells requires ATP7B. (**A**) IGROV (green) and IGROV-CP20 (red) cells were treated with the indicated concentrations of cisplatin for 24 or 48 h and viability of the cells was then evaluated using the MTT assay. (**B**,**F**,**I**) Western blots (and corresponding density quantifications) show ATP7B (B), ATP7A (F), and CTR1 (I) protein levels in IGROV and IGROV-CP20 cells (*n* = 3 experiments; * *p* < 0.05, ** *p* < 0.01, *t*-test). (**C**,**G**,**J**) qRT-PCR shows ATP7B (C), ATP7A (G), and CTR1 (J) mRNA levels in IGROV and IGROV-CP20 cells (*n* = 3 experiments; * *p* < 0.05, *t*-test). (**D**) IGROV and IGROV-CP20 cells were labeled for ATP7B and Golgin-97. Graphs (right column) indicate higher levels of ATP7B signal in IGROV-CP20 cells respect to the parental line. (**E**) IGROV-CP20 cells were transfected with control (siControl) or ATP7B-specific siRNAs and incubated with 50 µM cisplatin. MTT assay show reduced tolerance to cisplatin in ATP7B-silenced cells (*n* = 3 experiments; ** *p* < 0.01, ANOVA). (**H**) IGROV-CP20 cells were transfected with control (siControl) or ATP7A-specific siRNAs and incubated with 50 µM cisplatin. MTT assay did not detect viability differences between control and ATP7A-silenced cells upon cisplatin treatment. Scale bar: 25 µm (**D**).

**Figure 2 cancers-12-00608-f002:**
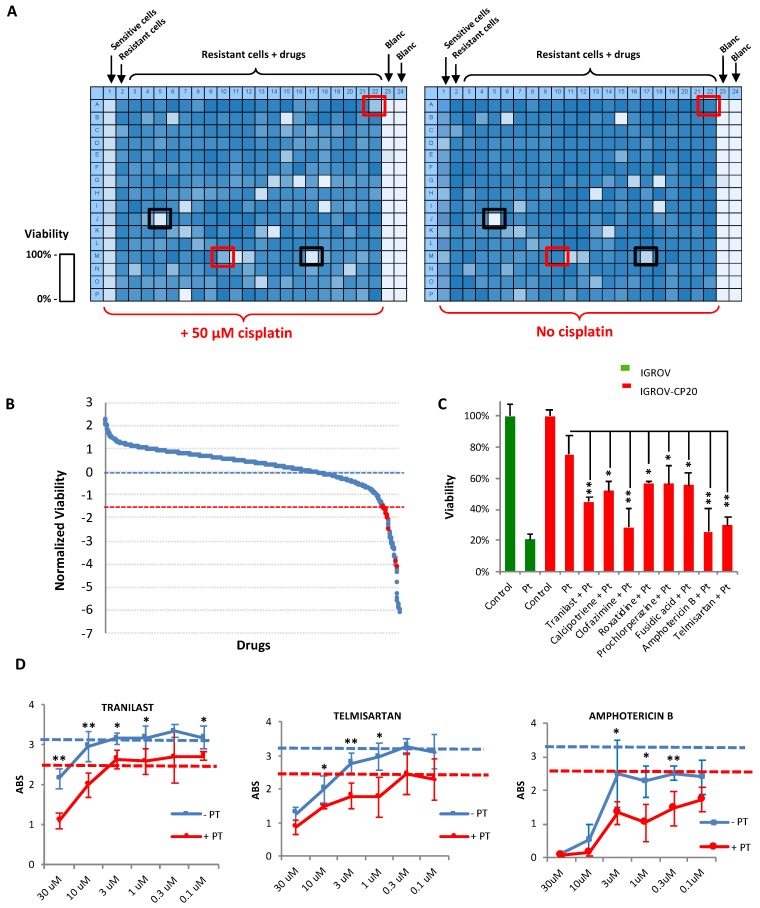
Synthetic lethality screening revealed FDA-approved drugs that promote platinum (Pt)-mediated death of IGROV-CP20 cells. (**A**) Heat maps showing the MTT signal (with blue indicating cell survival and white cell death) in two 384-well plates from the high-throughput screen (HTS). Sensitive IGROV cells were used as controls for cisplatin efficiency, while IGROV-CP20 cells were seeded in the central areas of each plate and treated with drugs from a library of 1280 FDA-approved molecules. For each set of drugs from the library one plate was treated with 50 µM cisplatin while the second plate with identical array of drugs was left without cisplatin. Drugs killing the cells only in combination with cisplatin were considered as the hits (red boxes), while drugs inducing cell death per se (black boxes) were excluded. (**B**) The panel shows the outcome of the screening. The average viability of IGROV-CP20 cells in cisplatin was considered as 0 (blue dash line). Standard deviation (SD) values of IGROV-CP20 cells in cisplatin were used to normalize the average MTT signal for each drug. Each drug, whose combination with cisplatin caused decrease in viability beyond an arbitrary threshold of 1.5 SDs (red dash line) was considered as a hit (red circle) if this drug did not induce toxicity without cisplatin. (**C**) The graph shows the percentage viability of IGROV-CP20 cells after treatment with the positive hits in combination with cisplatin (Pt) (*n* = 3 experiments; * *p* < 0.05, ** *p* < 0.01, ANOVA). (**D**) Quantification of MTT absorbance (as readout of viability) shows dose-response curves of Tranilast, Telmisartan, and Amphotericin B added to IGROV-CP20 cells with (red line) or without (blue line) 50 µM cisplatin. The blue dashed line shows viability in untreated cells, while the red dashed line shows viability in the cells treated with cisplatin alone. Data represent the average of six different experiments (* *p* < 0.05, ** *p* < 0.01, ANOVA).

**Figure 3 cancers-12-00608-f003:**
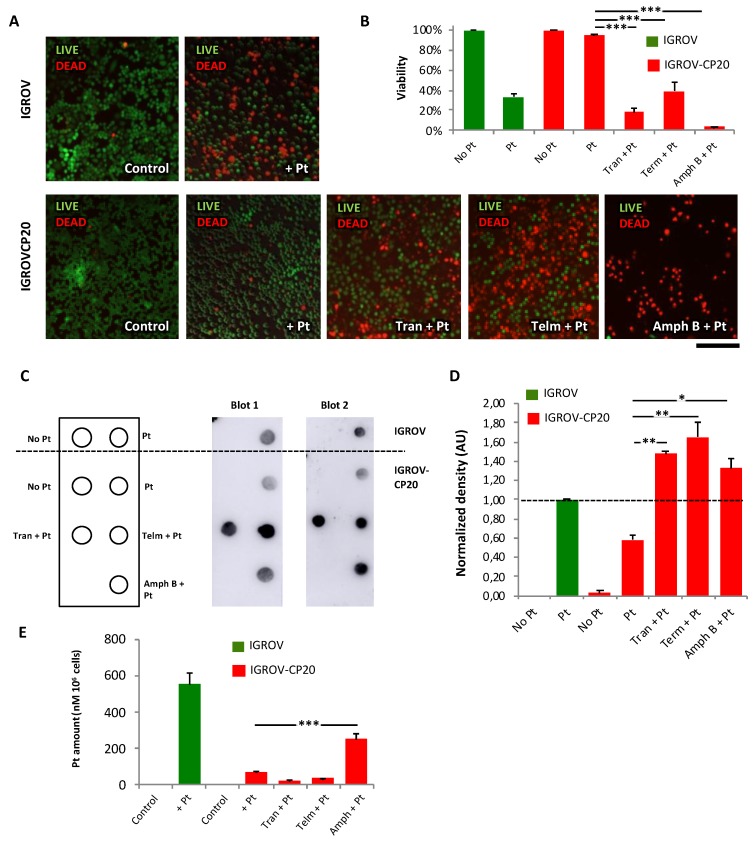
FDA-approved drug hits promote cisplatin toxicity and stimulate DNA adduct formation in resistant IGROV-CP20 tumor cells. (**A**) Representative images of live/dead assay (see Section 4) showing the effect of 50 µM cisplatin (Pt) on the cell viability of IGROV or IGROV-CP20 cells, as indicated. Pretreatment with 10 µM Tranilast, Telmisartan, or Amphotericin B for 24 h increased cell death of IGROV-CP20 cells when combined with cisplatin. (**B**) Quantification of experiments shown in panel A reveals a significant decrease in viability of IGROV-CP20 cells treated with a mixture of cisplatin and indicated drugs (*n* = 10 fields; *** *p* < 0.001, ANOVA). (**C**) Pt adducts were evaluated via dot immuno-blot (see Section 4) with DNA samples from IGROV cells (above the dashed line) and IGROV-CP20 cells (below the dashed line), which were spotted as indicated in the membrane map (on the left). Two dot blot images demonstrate that cisplatin (Pt) in association with Tranilast, Telmisartan, or Amphotericin B induces a significant increase in the DNA adduct signal compared to the IGROV-CP20 treated with cisplatin alone (**D**) The graph shows quantification of the DNA adduct signal in dot blot experiments (*n* = 10 fields; ** *p* < 0.01, * *p* < 0.05, ANOVA). (**E**) Cells were treated as in panel A and prepared for ICP-MS (see Section 4) to evaluate intracellular platinum levels. Only the combination of Amphotericin B with cisplatin led to an increase in the overall platinum levels in the cells. Sensitive IGROV cells (green bars) were used as a positive control for cisplatin treatment. The Pt content of each specimen was normalized to the total cell numbers as nM of PT in 10^6^ cells (*n* = 3 experiments; *** *p* < 0.001, ANOVA). Scale bar: 320 µM (**A**).

**Figure 4 cancers-12-00608-f004:**
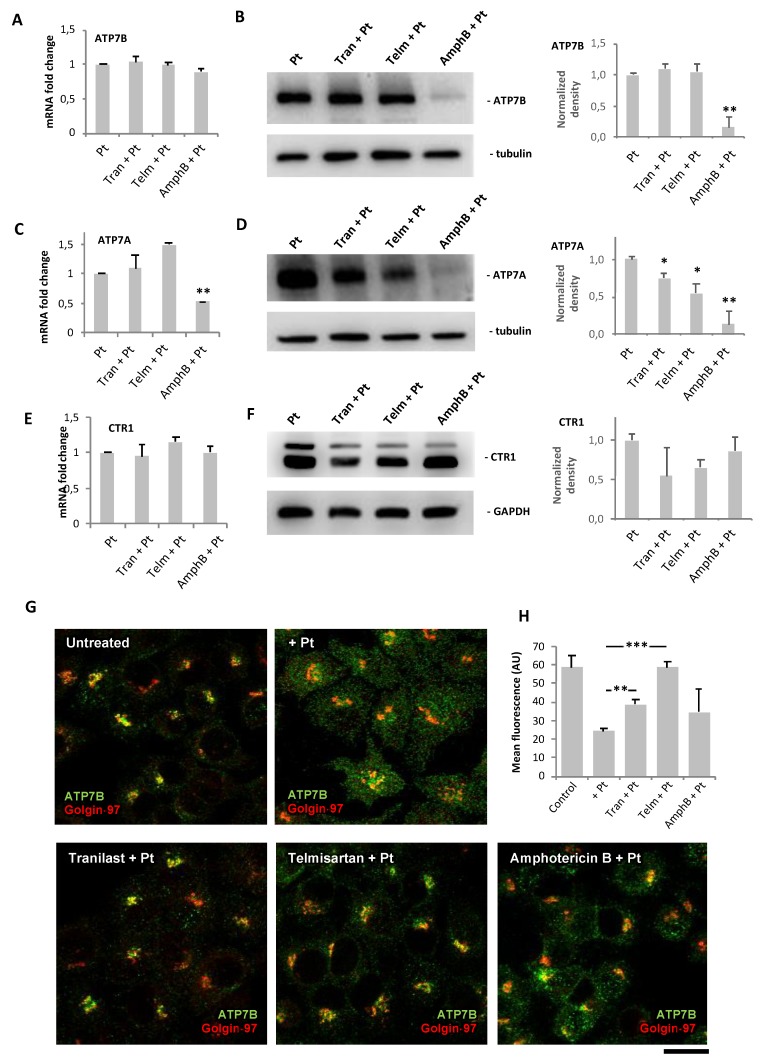
Impact of drug hits on expression of copper transporters and Pt-mediated trafficking of ATP7B in IGROV-CP20 cells. (**A**–**F**) IGROV-CP20 cells were treated with 10 µM Tranilast, Telmisartan, or Amphotericin B for 24 h and then 50 µM cisplatin was added for 24 h. The cells were then prepared for qRT-PCR (A,C,E) or Western blot ((B,D,F); see also signal density quantification in each panel) to evaluate the expression of ATP7B (A,B), ATP7A (C,D), and CTR1 (E,F) (for each qRT-PCR and blot *n* = 3 experiments; * *p* < 0.05, ** *p* < 0.01, ANOVA). (**G**,**H**) The cells were treated with 10 µM Tranilast, Telmisartan, or Amphotericin B for 24 h and then exposed to 50 µM cisplatin for 4 h to activate ATP7B trafficking. Confocal images (**G**) show that the drugs promote ATP7B retention in the Golgi (labeled with Golgin-97) upon cisplatin treatment as also revealed by quantification of the ATP7B signal (**H**) in the Golgi region (*n* = 10 fields; ** *p* < 0.01, *** *p* < 0.001, ANOVA). Scale bar: 15 µm (**G**).

**Figure 5 cancers-12-00608-f005:**
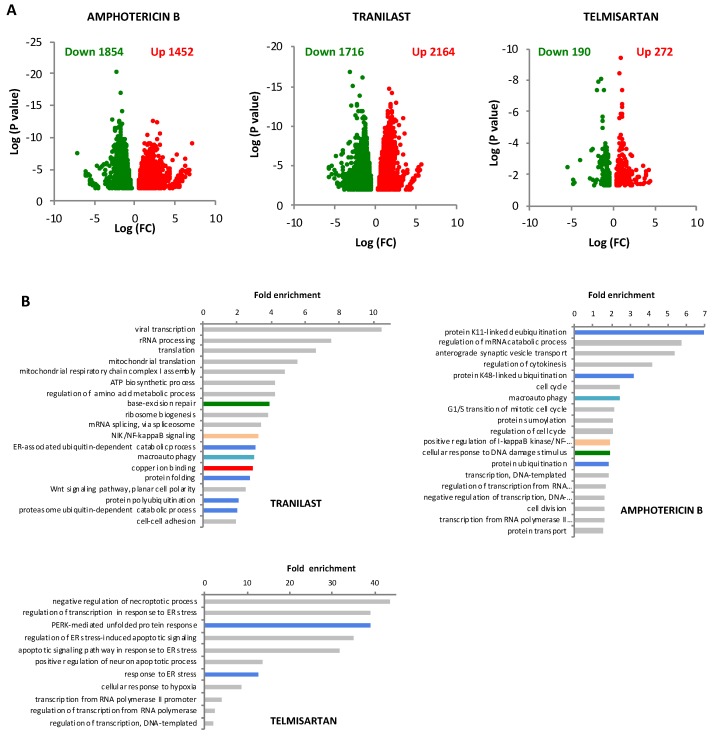
Impact of FDA-approved drugs on the transcriptome in cisplatin-treated IGROV-CP20 cells. (**A**) IGROV-CP20 cells were exposed for 24 h to 50 µM cisplatin either directly or 24 h after pretreatment with 10 µM Tranilast, Telmisartan, or Amphotericin B and prepared for QuantSeq analysis of mRNA (see Section 4). Volcano plot diagrams show down- and upregulated genes in IGROV-CP20 cells treated with the combination of the indicated drug and cisplatin compared to cisplatin alone. Data are expressed as Log of mRNA fold change. (**B**) The diagrams show GO enrichment analysis of genes whose expression was downregulated by Tranilast, Telmisartan, or Amphotericin B in cisplatin-treated cells. Pt-resistance pathways in common between at least two drugs are depicted with similar colors: DNA repair (green); protein quality control (blue); macroautophagy (cyan) and NFKB signaling (pink). Specific enrichment in downregulated copper ion binding genes (red bar) was detected only in Tranilast-treated cells.

**Figure 6 cancers-12-00608-f006:**
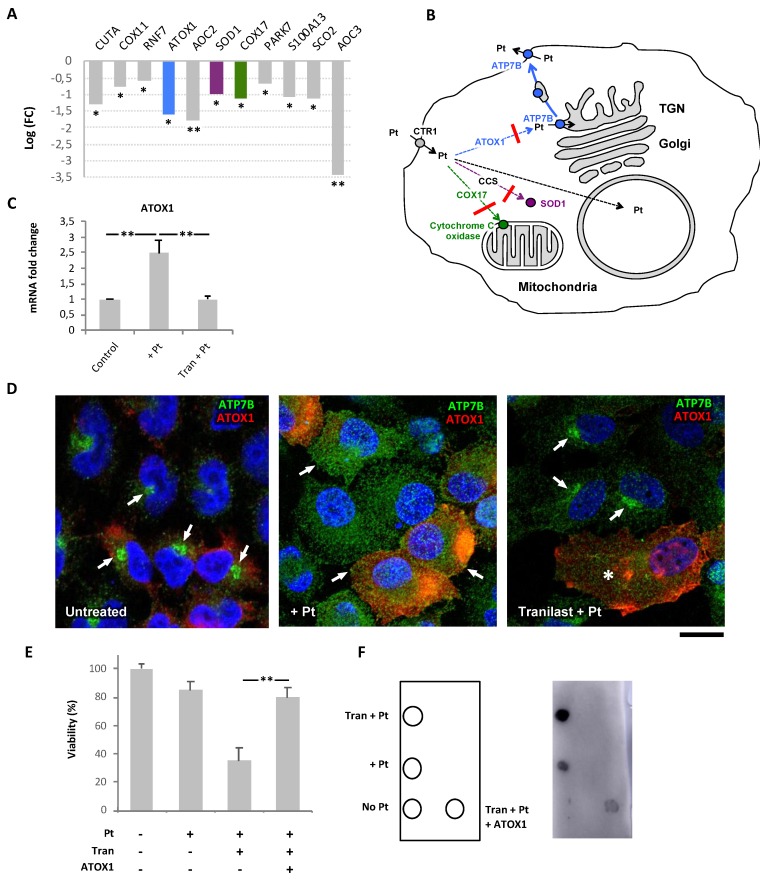
Impact of Tranilast on Pt-transporting pathways in cisplatin-resistant IGROV-CP20 cells. (**A**) QuantSeq analysis shows the impact of Tranilast on the expression of the copper ion binding genes in cisplatin-treated IGROV-CP20 cells (*n* = 3 experiments; * *p* < 0.05, ** *p* < 0.01, ANOVA). The genes playing a key role in the distribution of Pt across the cell (ATOX1, SOD1, and COX17) are shown in different colors corresponding to the specific intracellular route shown in panel B. (**B**) The scheme depicts Pt intracellular routes. Downregulation of ATOX1, SOD1, and COX17 by Tranilast (red crossbars) inhibits Pt delivery to the secretory pathway (blue arrows), mitochondria (green arrows), or SOD1-mediated detoxification pathway (magenta arrow). This favors delivery of Pt to the cell nucleus (dashed black arrow) leading to enhanced DNA damage. (**C**) qRT-PCR showed an increase in ATOX1 mRNA in cells treated with 50 µM cisplatin, while Tranilast inhibited Pt-mediated induction of ATOX1 expression (*n* = 3 experiments; ** *p* < 0.01, ANOVA). (**D**) IGROV-CP20 cells were transfected with pCDNA-ATOX1-FLAG and treated with 50 µM ciaplatin (Pt) or with a combination of 10 µM Tranilast and Pt (as indicated in the figure). The cells were then immuno-stained for ATP7B and FLAG. ATP7B was detected in the Golgi area in untreated cells (arrows, left panel) and at the cell surface and peripheral structures in Pt-treated cells (arrows, mid panel) regardless of ATOX1 overexpression. Tranilast blocked ATP7B in the Golgi area in Pt-treated cells, which did not overexpress ATOX1 (arrows, right panel), but failed to inhibit Pt-mediated redistribution of ATP7B from the Golgi in ATOX1-overexpressing cells (asterisks, right panel). (**E**) MTT viability assay was performed in cells treated with 50 µM cisplatin alone or in combination with 10 µM Tranilast (with and without ATOX1 overexpression). The plot shows that Tranilast did not reduce the viability of Pt-treated cells when ATOX1 is overexpressed (*n* = 3 experiments; ** *p* < 0.01, ANOVA). (**F**) Pt adducts were evaluated via dot immuno-blot of DNA samples spotted as indicated in the map on the left. Cisplatin (Pt) in association with Tranilast induced a significant increase in the DNA adduct signal compared to cells treated with cisplatin alone. ATOX1 overexpression inhibited the ability of Tranilast to promote DNA adduct formation in Pt-treated cells. Scale bar: 7 µm (**D**).

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
