# Peer review of "Synthetic Lethality Screening Identifies FDA-Approved Drugs that Overcome ATP7B-Mediated Tolerance of Tumor Cells to Cisplatin"

_cancers, 2020, doi:10.3390/cancers12030608_

Round 1

Reviewer 1 Report

The manuscript by Mariniello et al. uses a synthetic lethal screen with a FDA/EMA-approved drug library to identify mechanisms of resistance to Cisplatin mediated by ATP7B expression. The authors report three drugs that selectively increase cellular susceptibility to cisplatin:  Tranilast, Telmisartan and Amphotericin B. Furthermore, the authors identify mechanisms of action of these new drugs in the context of copper homeostasis either by increasing the uptake of cisplatin or by changing the expression and trafficking of ATP7 paralogues, or by modifying the expression of metallochaperones such as ATOX1.

The paper is a very important study, well-designed, and rigorously conducted. Results and conclusions are well balanced.

I do not have any substantive comments about the science which is excellent. However, I have some recommendations in the data presentation, statistics and some figure experiments that I would like the authors to address.

1) Figs 1 E and H. There is no documentation of the efficacy of the siRNA.

2) Fig. 2A, the color code bar for live-death is missing the color scale

3) Fig. 2B, the authors should report all drugs that significantly increase cell viability in the presence of cisplatin. This is important information.

4) Figs. 2C and 2D are missing statistics

5) Fig. 4 C is missing stats

6) Fig. 4B IF panels are mislabeled. It says AP7B instead of ATP7B

7) Fig. 5 needs a supplementary table with all the hits, their FC and their statistics

8) Fig. 6A needs stats.

9) Check for typos. For example, page 4 line 132 says “MTT calorimetric assay”. I should say “MTT colorimetric assay” 

Author Response

The paper is a very important study, well-designed, and rigorously conducted. Results and conclusions are well balanced.

We would like to thank the Reviewer for encouraging evaluation of the paper and very helpful comments.

1) Figs 1 E and H. There is no documentation of the efficacy of the siRNA.

RNAi efficiency for mentioned experiments was evaluated using qRT-PCR and now reported in Supplementary Fig. 1C, D.

2) Fig. 2A, the color code bar for live-death is missing the color scale

The figure was modified according Reviewer’s suggestion with color code bar showing 0 to 100% cell viability.

3) Fig. 2B, the authors should report all drugs that significantly increase cell viability in the presence of cisplatin. This is important information.

We agree with Reviewer. The drugs, which increase resistance, might represent a risk factor during cisplatin chemotherapy. We listed them in Supplementary Fig. 3B and mentioned in the text (lines 155-157).

4) Figs. 2C and 2D are missing statistics. 5) Fig. 4 C is missing stats. 8) Fig. 6A needs stats.

Statistics was added to all figures.

6) Fig. 4B IF panels are mislabeled. It says AP7B instead of ATP7B

Labels in these panels were corrected to ATP7B

7) Fig. 5 needs a supplementary table with all the hits, their FC and their statistics

Making a single supplementary table with all data for 7-8 thousands of genes would be difficult. Therefore, we added a big supplementary dataset Exel file with 3 spreadsheets (one for each drug) containing all down- and up-regulated genes, their FCs and statistics.

9) Check for typos.

Corrections were made according the Reviewer’s suggestion.

Reviewer 2 Report

The paper entitled "Synthetic lethality screening identifies FDA-approved drugs that overcome ATP7B-mediated tolerance of tumor cells to cisplatin" describe the identification of three FDA approved small molecules that could overcome platinum resistance in certain cells. The three hits were identifies using an interestin HTS approach which was correctly validated.

The results present herein, could be of extreme interest for the scientific community and highlight the importance of identify new molecular targets for FDA approved small molecules which could speed-up and reduce the cost of the drug discovery process.

On other hand, a new compound based on Cis-Platin and Tranilast was described in 2018 in ChemmComm (10.1039/C8CC02071J) and the results presented in this paper could strengthen some of the observations of Mariniello et al. and therefore should be cited in the discussion.

Overall, an interesting paper that with some adjustments I recommend for publication in Cancers.

Author Response

The results present herein, could be of extreme interest for the scientific community and highlight the importance of identify new molecular targets for FDA approved small molecules which could speed-up and reduce the cost of the drug discovery process.

On other hand, a new compound based on Cis-Platin and Tranilast was described in 2018 in ChemmComm (10.1039/C8CC02071J) and the results presented in this paper could strengthen some of the observations of Mariniello et al. and therefore should be cited in the discussion.

 We would like to thank Reviewer for the positive remarks and helpful suggestion. We cited the ChemmComm publication in the discussion section of the revised manuscript (lines 357-358).

Reviewer 3 Report

The manuscript “Synthetic lethality screening identifies FDA approved drugs that overcome ATP7B-mediated tolerance of tumor cells to cisplatin” by Mariniello et al addresses a relevant clinical question on platinum resistance in ovarian cancer.The paper is well structured and the conclusion has been supported by the reported results.

There are some comments on the experimental design:

  • The authors showed that among the three screened drugs the Amphotericin B induced the highest reduction of ATPP7A and ATP7B levels along with the highest reduction of cell viability in resistant cells. Furthermore, they screened the effects of the selected drugs also on the expression of other genes involved in different Pt-resistance pathways and they found significant differences between the different treatments in modulation of such genes. The authors concluded that the Amphotericin B remain the most effective drug but it appears to be the least safe due to its high toxicity, which limit its clinical use. Did the author try to perform cell viability assays combining Telmisartan or Tranilast with low dose of Amphotericin B to better improve tumor response to cisplatin?

Author Response

The manuscript “Synthetic lethality screening identifies FDA approved drugs that overcome ATP7B-mediated tolerance of tumor cells to cisplatin” by Mariniello et al addresses a relevant clinical question on platinum resistance in ovarian cancer. The paper is well structured and the conclusion has been supported by the reported results.

 We would like to thank the Reviewer for the positive evaluation of our manuscript and helpful comments.

The authors concluded that the Amphotericin B remain the most effective drug but it appears to be the least safe due to its high toxicity, which limit its clinical use. Did the author try to perform cell viability assays combining Telmisartan or Tranilast with low dose of Amphotericin B to better improve tumor response to cisplatin?

 We tried to combine lower doses of Amphotericin B with either Telmisartan or Tranilast. It was, however, quite difficult to observe any statistically significant synergistic impact on cisplatin cytotoxicity. For this reason, we did not include these finding in the manuscript. Usually effectiveness of Amphotericin B at the concentration above 0.3 uM was not improved by Telmisartan or Tranilast, while concentrations of Amphotericin B below 0.3 uM did not impact the magnitude of response of Pt-treated cell to Telmisartan or Tranilast.